# Real-Time (iOCT) Guided Epiretinal Membrane Surgery Using a Novel Forceps with Laser-Ablated Microstructure Tip Surface

**Agharza Ashurov [1,*], Argyrios Chronopoulos [1], Julia Heim [1], James Scott Schutz [1], Carl Arndt [2] and Lars-Olof Hattenbach [1]**

[1] Department of Ophthalmology, Hospital of Ludwigshafen, Teaching Hospital of the University of Johannes Gutenberg-University Mainz and Medical Faculty of the University of Mannheim, 67063 Ludwigshafen, Germany

[2] Department of Ophthalmology, University Hospital of Reims, 51100 Reims, France

* Correspondence: a.ashurov@gmx.de; Tel.: +49-0621-503-3058; Fax: +49-0621-503-770030

**Abstract:** Purpose: We investigated intraoperative OCT (iOCT)—guided epiretinal membrane (ERM) and internal limiting membrane (ILM) removal using a novel forceps with a laser-ablated tip surface; it was designed to help prevent indentation force, shear stress, or tractional trauma when grasping very fine membranes. Patients and Methods: This retrospective study included patients who underwent 23- and 25-gauge pars plana vitrectomy (PPV) for vitreoretinal interface disorders. ERM and ILM peeling was performed under guidance with microscope-integrated iOCT using novel ILM forceps with laser-ablated tip surfaces. These forceps were engineered to enhance friction when grasping tissue. Evaluation of ERM/ILM manipulation included postoperative slow-motion video analysis of the number of grasping attempts, initial ILM mobilization, and observed damage to retinal tissue. Results: ERM/ILM removal was successfully performed in all patients, with an average of four grasp actions to initial membrane mobilization (91%). Additional use of a diamond-dusted membrane scraper was used in two cases (9%). Mean best-recorded visual acuity (BRVA) logMAR improved from $0.5 \pm 0.34$ to $0.33 \pm 0.36$ ($p = 0.05$) and mean central retinal thickness (CRT) improved from $462 \pm 146$ μm to $359 \pm 78$ μm ($p = 0.002$). Postoperative iOCT video analysis demonstrated hyper-reflectivity of the inner retinal layers associated with retinal hemorrhage in five eyes (22%), but no grasping-related retinal breaks. Conclusions: The texturized surface on the tips of the ILM forceps were found to be helpful for mobilizing ILM edges from the retinal surface. iOCT-guided ERM surgery also allowed for improved intraoperative tissue visualization. We believe that these two technologies helped reduce both unnecessary surgical maneuvers and retinal damage.

**Keywords:** internal limiting membrane surgery; epiretinal membrane surgery; intraoperative optical coherence tomography; vitreoretinal interface; novel forceps

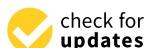

## 1. Introduction

Optical coherence tomography is an indispensable tool in modern diagnosis of vitreoretinal disorders [1]. The introduction of spectral domain optical coherence tomography (SD-OCT) has provided vitreoretinal surgeons with higher-quality retinal scans enabling better diagnostics and surgical outcomes [2]. The development of intraoperative microscope-mounted OCT (iOCT) was a natural extension following the indispensability of OCT and the growing demand for improved tissue visualization during vitreoretinal microsurgical procedures [3–15]. Since the first description by Dayani et al., different study groups have presented iOCT setups including microscope-integrated SD-OCT [4]. The latter provides the surgeon with improved vitreoretinal interface visualization during surgery and instant assessment of microanatomical changes associated with surgical manipulation of the retina [3–15].

In recent years, it has become evident that vitreoretinal interface manipulation leads to inner retinal abnormalities with iatrogenic retinal injuries including retinal holes or breaks,

vascular leakage, and nerve fiber layer atrophy [16–18]. These defects are probably the result of tractional/shear stress transmitted into the retina [19,20]. Surgical instrument design is important in optimal vitreoretinal interface surgery to reduce shear stress and tractional forces into the retina which may cause retinal damage [8,17]. The design elements of the forceps are intended to support atraumatic initiation of the ILM peel and mitigate tearing of the membrane during delamination [17]. The laser-ablated microsurface of the Sharkskin ILM forceps (Figure 1) has membrane scraper-like increased friction built into the forceps tips. ERM and ILM peeling were reviewed with the Sharkskin ILM forceps using slow-motion-mode iOCT video. A review was performed to determine whether indentation force or physical trauma was prevented when grasping very fine membranes, as well as to evaluate the retinal change with iOCT following the membrane peeling.

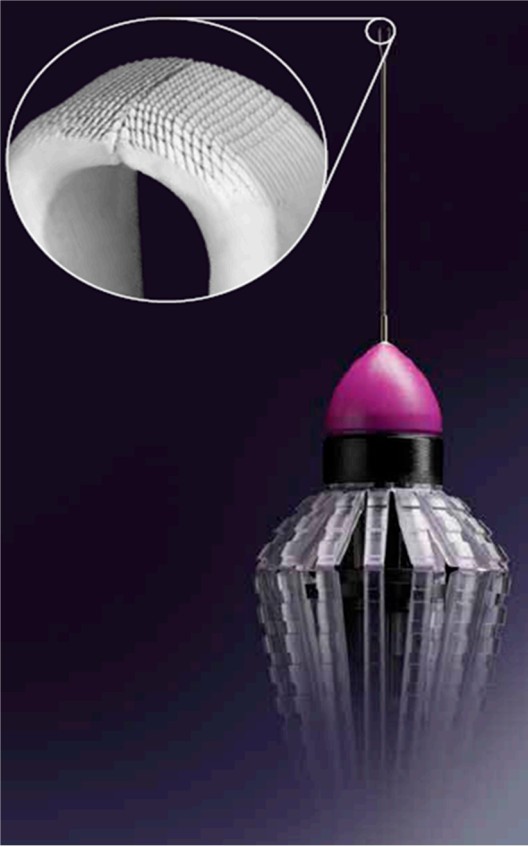

**Figure 1.** Finesse Sharkskin Forceps, Alcon, Ft. Worth, TX, USA. A laser-ablated micro-structured tip surface and larger platform designed for precision grasping during ILM peeling and enhanced friction between tissue and the forceps tip.

## 2. Materials and Methods

This retrospective study included 23 eyes of 23 patients. A standard 3-port pars plana vitrectomy with a Chandelier light source was performed using the Constellation 23- and 25-gauge vitrectomy system (Alcon Laboratories Inc, Fort Worth, TX, USA). After vitrectomy, the posterior vitreoretinal interface was visualized by iOCT (OPMI LUMERA® 700, ZEISS© with integrated OCT). The ILM was stained using 0.1 mL of Brilliant Blue G (Brilliant Peel; Geuder, Germany) for approximately 1 min after stopping infusion and then excess dye was removed. iOCT was used to identify small discontinuities/gaps between the ERM and the nerve fiber layer (Figure 2), which were used as a scaffold to initiate membrane removal in a targeted manner with minimal nerve fiber layer disturbance. The ILM was grasped with end-gripping forceps at a thick or wrinkled spot and a horizontal ILM strip was peeled off (Figure 3). Fluid–air exchange was performed using a 23- and

25-gauge flute needle held nasal to the disk. The vitreous cavity was then filled either with air or 25% perfluoropropane gas.

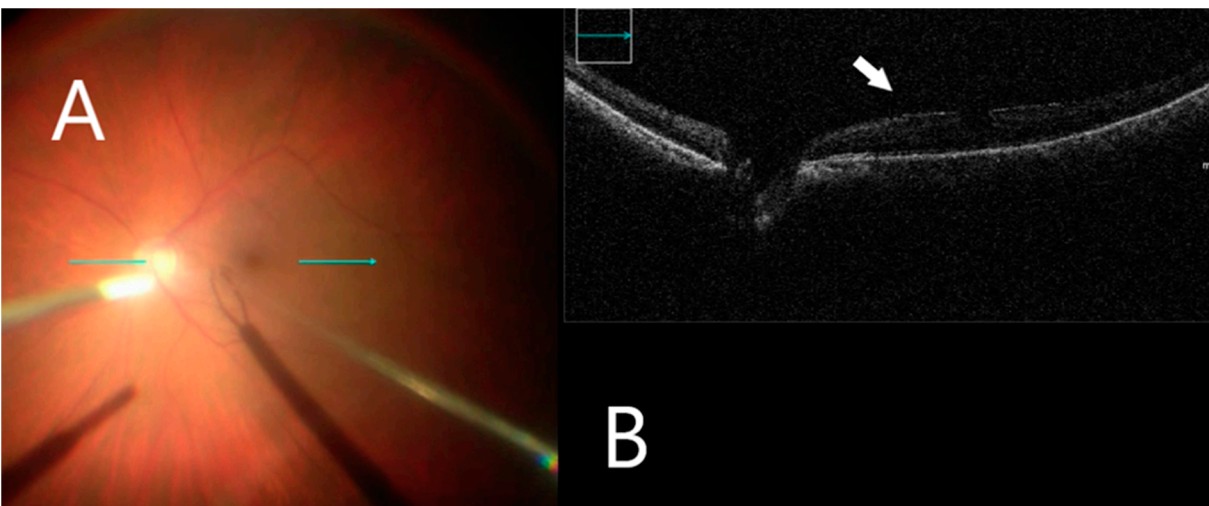

**Figure 2.** iOCT immediately before membrane mobilization. (**A**) Intraoperative surgical view before membrane peeling. Horizontal scan line (green arrow). (**B**) Identifying small discontinuations between ERM and nerve fiber layer (white arrow).

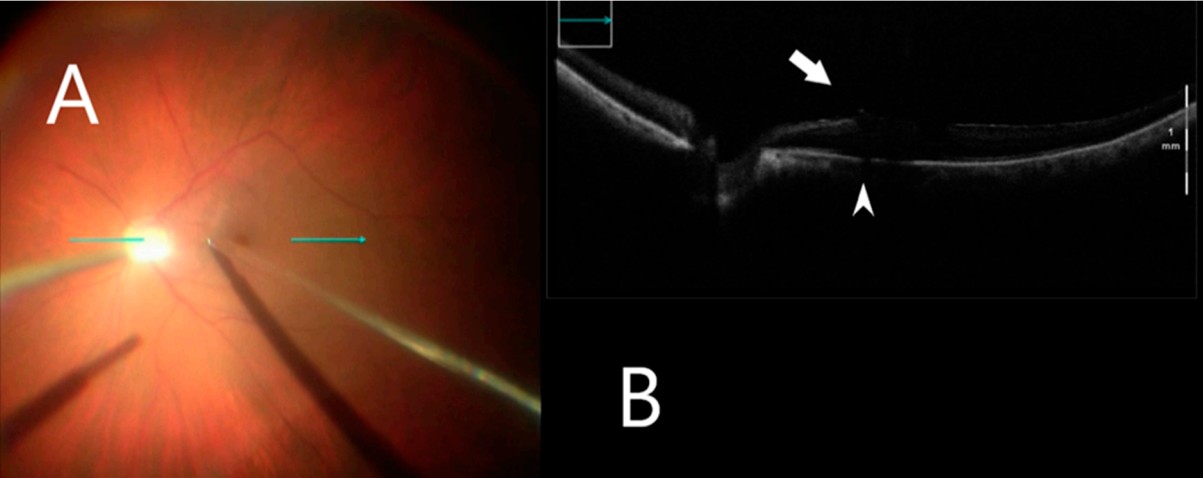

**Figure 3.** iOCT during membrane peeling. (**A**) Intraoperative surgical view during membrane peeling. Horizontal scan line (green arrow). (**B**) Grasping an optimal point of ILM using iOCT (white arrow). B-scan of the posterior pole shows shadowing artifact posterior to the metallic membrane peeling (white arrowhead).

*2.1. Data and Statistical Analysis*

Evaluation of ERM/ILM manipulation included postoperative slow-motion mode iOCT video analysis with regard to the number of grasping attempts to initial ILM mobilization and detection of damage to retinal tissue. The additional use of a diamond-dusted membrane scraper for ERM/ILM mobilization was recorded. BRVA and CRT on OCT were assessed preoperatively and at 6 months postoperatively. BRVA was converted to the logarithm of the minimum angle of resolution (LogMAR) for statistical analysis. BRVA in LogMAR and mean CRT were evaluated against baseline for each time-point unpaired *t*-test for parametric data. The *p*-values were calculated, with a value of less than 0.05 indicating statistical significance.

*2.2. Intraoperative OCT Analysis*

Surgical video and iOCT video files were analyzed for the specific anatomic location of the membrane mobilization at three surgical points: immediately before membrane peeling, after initial membrane mobilization, and after complete membrane removal. Before membrane peeling, iOCTs were reviewed for any pre-existing retinal abnormalities. After membrane mobilization and complete membrane removal, iOCTs were reviewed to evaluate intraoperative retinal alterations to the retina such as increased subretinal hyporeflectance, retinal break development, increased inner retinal hyper-reflectivity with or without retinal hemorrhage.

**3. Results**

*3.1. Demographics*

Twenty-three eyes of 23 patients (14 men and 9 women) underwent 25 g or 23 g pars plana vitrectomy with the following indications: idiopathic macular pucker (11), full-thickness macular hole (7), epiretinal membrane after vitrectomy for retinal detachment (3), vitreomacular traction syndrome (2) between 2019 and 2020 (Table 1). The mean age of patients was 75 years, eye laterality was similar (right eye 14, left eye 19) and all eyes were pseudophakic (Table 1). The macular hole was closed successfully in all 7 eyes of 7 patients with a single surgery.

**Table 1.** Clinical Demographics.

| Diagnosis Group | ERM, $n = 11$ | FTMH, $n = 7$ | VMT, $n = 2$ | ERM after RD, $n = 3$ | All, $n = 23$ |
|---|---|---|---|---|---|
| Sex, *n* | | | | | |
| Men | 6 (55%) | 4 (57%) | 1 (50%) | 3 (100%) | 14 (61%) |
| Women | 5 (45%) | 3 (43%) | 1 (50%) | 0 | 9 (39%) |
| Age, years | 71 | 74 | 80 | 73 | 75 |
| Laterality of the eye, *n* | | | | | |
| Right | 7 (64%) | 4 (57%) | 1 (50%) | 2 (67%) | 14 (61%) |
| Left | 4 (36%) | 3 (43%) | 1 (50%) | 1 (33%) | 9 (39%) |
| Lens status, *n* | | | | | |
| Phakic | 0 | 0 | 0 | 0 | 0 |
| Pseudophakic | 11 (100%) | 7 (100%) | 2 (100%) | 3 (100%) | 23 (100%) |

*3.2. Central Retinal Thickness*

Mean preoperative CRT was 462 μm with a range from 251 μm to 911 μm compared to mean final postoperative CRT of 359 μm with a range from 283 μm to 637 μm. Postoperative SD-OCT, performed on all patients, on average, at the 6-month follow-up, showed CRT decreased by an average of 23% ($p = 0.002$, Student's unpaired *t*-test).

*3.3. Vision*

Mean pre-operative logMAR BRVA (Table 2) was $0.5 \pm 0.34$ with a range from 1.3 to 0. Mean final postoperative BRVA was $0.33 \pm 0.36$ with a range from 1 to 0. BRVA improved an average of 34% ($p = 0.04$ Student's unpaired *t*-test) at 6 months after surgery.

**Table 2.** Retinal Alterations Identified with Intraoperative OCT.

| Diagnosis Group | ERM, *n* = 11 | FTMH, *n* = 7 | VMT, *n* = 2 | ERM after RD, *n* = 3 | All, *n* = 23 |
|---|---|---|---|---|---|
| BRVA logMAR, mean ± SD before Peeling | 0.52 ± 0.23 | 0.69 ± 0.4 | 0.3 ± 0.14 | 0.8 ± 0.26 | 0.5 ± 0.34 |
| BRVA logMAR, mean ± SD after Peeling | 0.27 ± 0.14 | 0.43 ± 0.56 | 0.6 ± 0.57 | 0.13 ± 0.15 | 0.33 ± 0.36 |
| Subretinal hyporeflectance | 2 (18%) | 0 | 1 (50%) | 1 (33%) | 4 (17%) |
| Retinal break | 0 | 0 | 0 | 0 | 0 |
| Increased inner retinal hyper-reflectivity associated with retinal hemorrhage | 2 (18%) | 1 (14%) | 1 (50%) | 1 (33%) | 5 (22%) |
| Increased inner retinal hyper-reflectivity not associated with retinal hemorrhage | 6 (55%) | 5 (71%) | 1 (50%) | 2 (66%) | 14 (61%) |

### 3.4. Intraoperative Image Assessment

Seventeen percent (17%) of all the cases (4 eyes) in the study had subretinal hyporeflectance noted on intraoperative imaging post-ILM peel. Immediately after ILM peeling, focal retinal hemorrhages associated with increased inner retinal hyper-reflectivity were noted within the area the membrane peeling in five eyes (22%) (Figure 4). An additional 14 eyes (61%) had increased inner retinal hyper-reflectivity unassociated with retinal hemorrhage (Figure 5). ERM/ILM removal was successfully performed in all patients with an average of four grasp actions (1 to 5 grasps) for initial membrane mobilization (91%). In two cases (9%), additional use of a diamond-dusted membrane scraper was needed. No grasping-related retinal holes or breaks were observed from multiple grasping attempts.

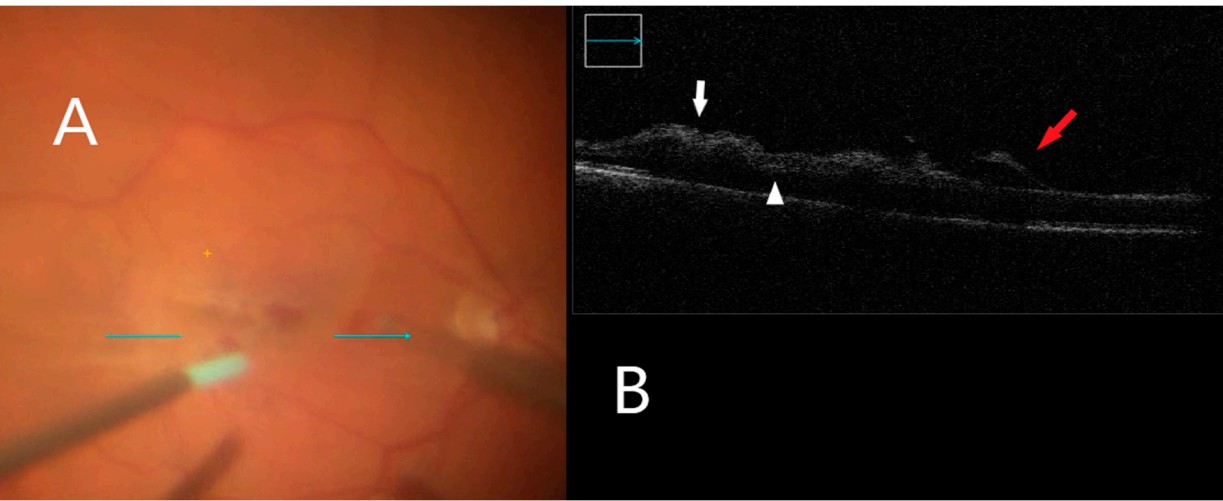

**Figure 4.** iOCT during membrane peeling. (**A**) Intraoperative surgical view during membrane peeling. Horizontal scan line (green arrow). (**B**) Increased inner retinal hyperreflectivity associated with retinal hemorrhage can be seen (white arrow). Subretinal fluid accumulation is also visible (white arrowhead). Lifted ILM is evident (red arrow).

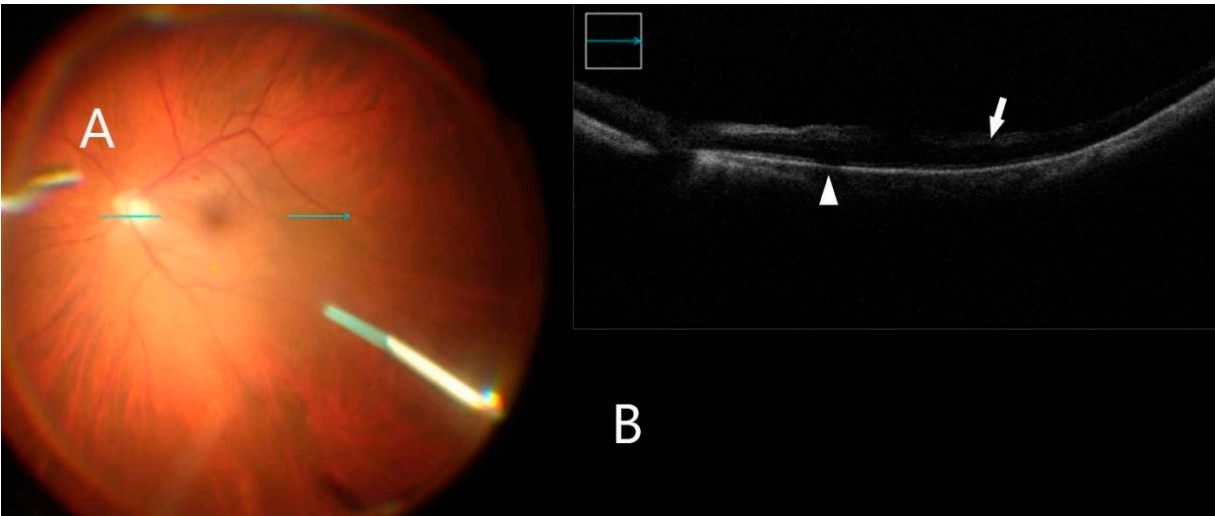

**Figure 5.** iOCT immediately after membrane peel. (**A**) Intraoperative surgical view after membrane peeling. Horizontal scan line (green arrow). (**B**) B-scan shows inner retinal hyper-reflectivity not associated with retinal hemorrhage (white arrow). Subretinal fluid accumulation is also visible (white arrowhead).

## 4. Discussion

In this study, we reported the intraoperative retinal changes identified with iOCT for vitreomacular interface disorders with using novel forceps featuring laser-ablated tip surface. Peeling of the ILM was enabled with an average of four grasps for initial membrane mobilization (91%) with only two cases requiring additional use of a diamond-dusted membrane scraper (9%). Four out of twenty-three eyes (17%) demonstrated subretinal hyporeflectance, but none of these four eyes showed clear subretinal fluid on iOCT. No retinal breaks with ERM/ILM removal occurred in the 23 study eyes.

OCT is very important for the preoperative diagnostic clinical management of vitre-oretinal interface disorders. The use of OCT for live intraoperative surgery has similar potential to enhance understanding of the clinical architectural change and provide the impact of surgical maneuvers on the retina [17,21,22]. Previous studies have suggested that iOCT may identify clinical alterations in the architecture of the retina, including in retinal detachment repair, ERM peeling, and macular hole (MH) repair [17,21,22]. The retinal changes appear in the photoreceptor layers after membrane peeling, an expansion of the outer retinal (OR) layer to retinal pigment epithelium (RPE) heights [17,21,22]. The expansion of the OR-RPE may reflect clear subretinal fluid, inner retinal hyper-reflectivity, or perhaps partial disinsertion from the RPE [17,21].

Membrane peeling with forceps and/or a diamond-duster membrane scraper is known to induce retinal alterations with subretinal hyporeflectance incidence at 9–28% after ILM peeling [17,21]. In the current report, subretinal hyporeflectance was observed in four eyes (17%). Directly following complete membrane removal, follow-up imaging con-firmed resolution of this hyporeflectance. The clinical significance of these intraoperative anatomical alterations are not completely understood but they might represent subclinical neurosensory retinal detachment or photoreceptor stretching [14,21,22].

Superficial retinal hemorrhages have been reported in 66% to 75% of ILM peeling using either a diamond-dusted membrane scraper or forceps [21,23,24]. Such hemorrhages are usually self-limited [21,24,25]. Uchida et al. demonstrated retinal hemorrhage-associated inner retinal hyper-reflectivity in 29% of patients with ILM peeled with only forceps and in 74%, associated inner retinal hyper-reflectivity without hemorrhage [21]. In the current report, inner retinal hyper-reflectivity with retinal hemorrhages was observed in 5 of 23 eyes (21%) and 14 of 23 eyes (61%) without hemorrhages. Etiologies for inner retinal hemorrhage

following the ILM elevation include direct trauma to the inner retina from the membrane mobilization or applied traction from the ILM when removed from the retinal surface.

Pavlidis et al. found significantly better visual acuity improvement after membrane peeling with no foveal contact as compared to that with foveal contact [26]. The higher the elevation of preretinal gliosis, the fewer grasping attempts were needed with a range of 1–4 grasps [26]. The current study demonstrated 91% (21) cases of successful membrane performed with just a Finesse Sharkskin™ ILM forceps with one to five grasp actions required for initial membrane mobilization.

Some surgeons use a diamond-dusted membrane scraper to both initiate and complete peeling [18,23]. It has been reported that less nerve fiber layer damage and retinal debris on transmission electron microscopy occurred during ILM peeling with forceps compared to a diamond-dusted membrane scraper technique [23]. Many surgeons use a direct "pinch" technique using custom-designed forceps to initiate a flap, attempting to avoid retinal tissue in the initial pinch. The membrane is grasped in forceps, lifted very slightly form the retinal surface, and then pulled tangentially, creating a flap with a rip point 180° from the direction of pull [27].

Novel instruments such as the Sharkskin ILM forceps designed to improve initial grasping to create an edge in the ILM may help minimize retinal damage. Validation of this observation needs to be confirmed by further studies.

## 5. Conclusions

Our study has some limitations: the limited number of patients and its retrospective nature. In conclusion, suitable instruments, such as the Sharkskin ILM forceps, allow for intraoperative real-time OCT guided micrometer-level epiretinal membrane removal and monitoring. These findings need to be confirmed with further comparative studies.

**Author Contributions:** Conceptualization, A.A., A.C., J.S.S. and L.-O.H.; methodology, A.A. and A.C.; software, A.C. and J.S.S.; validation, A.A., A.C. and J.S.S.; formal analysis, A.A. and A.C.; investigation, A.A., A.C. und J.H.; resources, J.H. and A.C.; data curation, A.A., J.H.; writing—original draft preparation, A.A.; writing—review and editing, A.A., A.C., J.S.S., C.A., L.-O.H.; visualization, A.A., A.C. and L.-O.H.; supervision, A.A., A.C. and L.-O.H.; project administration, A.A., L.-O.H.; funding acquisition, A.A. and L.-O.H. All authors have read and agreed to the published version of the manuscript.

**Funding:** A.A., A.C., J.H., J.S.S., C.A., L.-O.H. have non-financial support from Alcon during the conduct of the study; personal fees from Alcon; personal fees from Bausch and Lomb and personal fees from Carl Zeiss Meditec were received outside of the submitted work. Data analysis and manuscript preparation was supported by an Alcon grant (IIT # 63465423).

**Institutional Review Board Statement:** As the study was retrospective. The study was conducted in accordance with the Declaration of Helsinki and approved by the Ethics Committee of Landesärztekammer Rheinland-Pfalz (protocol code 2021-15782 and date of approval 26 August 2021).

**Informed Consent Statement:** Patient consent was waived due to the retrospective nature of the study.

**Data Availability Statement:** All data were fully anonymized and are available upon request.

**Conflicts of Interest:** The authors declare no conflict of interest.

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
