# Peer review of "Real-Time (iOCT) Guided Epiretinal Membrane Surgery Using a Novel Forceps with Laser-Ablated Microstructure Tip Surface"

_clinpract, doi:10.3390/clinpract12050086_

Round 1

Reviewer 1 Report

Ashurov et al present a study of 23 eyes who underwent membrane peeling for various indications using the sharkskin forceps from Alcon, who the authors receive personal fees from. The authors find that these 23 patients have improved BRVA (20/60 to 20/40 in Snellen) and improved retinal thickness after surgery. On iOCT analysis, their cohort underwent an average of 4 initial grasps causing subretinal hyporeflectance in 17% of patients, retinal hemorrhage in 21%, and 0 retinal breaks. The authors conclude that the novel forceps are “safer” in the summary statement and allow “reduced unnecessary retinal damage” in the conclusions of the abstract.

Major Concern

1. My primary concern with this study is that conclusion that the forceps is “safer” and “reduced” retinal damage are not supported by the data. The authors do not have a control group, so rightly go to the literature. They found subretinal hyporeflectance: 9%-28% vs 17% in their study, retinal hemorrhage 29% vs 21% in their study, and 1-4 average initial grasps vs 1-5 grasps in their study. All of these data support that the sharkskin forcep is safe and equivalent to prior outcomes. There is no data to support a benefit or that it “reduced unnecessary retinal damage”. No data is presented or discussed to compare inner retinal hyperreflectivity in 61% of patients. Zero breaks is an important finding, but breaks are rare and 23 patients is underpowered to detect a change.

2. The statistical analysis is confusing and lacks detail. The methods state that unpaired analysis was performed, but the text states a paired analysis was performed. Second, the methods state that baseline, 3 months, and 6 months were evaluated but the data only includes baseline and post-operative data points. Is this 3 months or 6 months? Last observation carried forward? The ideal analysis would be a repeated measures ANOVA comparing baseline, 3 months, and 6 months.

3. Fig 4B and 5B are not adequate to display any retinal hyperreflectivities, please enhance brightness and/or contrast

Minor comments:

1. Is this a prospective or retrospective study?

2. CRT in a FTMH case: is this a relevant comparison? Technically it is 0 at baseline. I would omit FTMH from CRT analysis. I would add percent FTMH closure with a single surgery.

3. Please define BRVA, ERM, ILM, CRT at first mention

Author Response

Dear Review,

Thank you for your fast answer and the very professionaly review of my manuskript. The points you noticed have been edited. Thank you for your comments.

  1. My primary concern with this study is that conclusion that the forceps is “safer” and “reduced” retinal damage are not supported by the data. The authors do not have a control group, so rightly go to the literature. They found subretinal hyporeflectance: 9%-28% vs 17% in their study, retinal hemorrhage 29% vs 21% in their study, and 1-4 average initial grasps vs 1-5 grasps in their study. All of these data support that the sharkskin forcep is safe and equivalent to prior outcomes. There is no data to support a benefit or that it “reduced unnecessary retinal damage”. No data is presented or discussed to compare inner retinal hyperreflectivity in 61% of patients. Zero breaks is an important finding, but breaks are rare and 23 patients is underpowered to detect a change.

Response: We corrected the word safer to safe, that’s what our data shows. We didn’t show „reduced“ damage but we commented that it may help reduce damage though the small number of paptients. This is our finding. We added the comparison of inner retinal hyperreflectivity in 61% of our patients.

  1. The statistical analysis is confusing and lacks detail. The methods state that unpaired analysis was performed, but the text states a paired analysis was performed. Second, the methods state that baseline, 3 months, and 6 months were evaluated but the data only includes baseline and post-operative data points. Is this 3 months or 6 months? Last observation carried forward? The ideal analysis would be a repeated measures ANOVA comparing baseline, 3 months, and 6 months.

Response: We have baseline and 6 months of data of all patients and 3 months data of not all patients. This is an unpaired analysis and has been corrected. 

  1. Fig 4B and 5B are not adequate to display any retinal hyperreflectivities, please enhance brightness and/or contrast

Response: I will send you high quality 4b und 5b pictures (see attachment).

Minor comments:

  1. Is this a prospective or retrospective study?

Response: Yes, This is a retrospective study.

  1. CRT in a FTMH case: is this a relevant comparison? Technically it is 0 at baseline. I would omit FTMH from CRT analysis. I would add percent FTMH closure with a single surgery.

Response: I will send you OCT-pictures, CRT before and after peeling by FTMH (see attachment).

  1. Please define BRVA, ERM, ILM, CRT at first mention

Response: BRVA, ERM, ILM, CRT were defined at the first mention.

best regards 

Agharza Ashurov

Reviewer 2 Report

This is an interesting study to report the results using a new design of an ILM forceps. Ideally this should be done in a comparative study which limits the conclusions of the current pilot study.

Define BRVA and CRT in abstract and text before use as well as any other abbreviations.

In abstract add logmar to brva. Also the postoperative vision is different in abstract and text.

Add the SD of BRVA in text.

Please explain further why was a diamond dusted scraper needed in two cases (more adherent more friable membrane etc?)

Please add a video for one of the operations to review.

Was there any post op complications such as retinal detachment or recurrent ERM?

In disclosure section you have to mention the disclosures of each author separately.

In discussion you mention that the number of grasps to remove the whole ERM by Mylonas et al was reported as 71 ±35.2 compared to a mean of 4 grasps in your study but that was for initial mobilisation. This is not comparable. Please indicate that.

You have no conclusions in the conclusions section.

Author Response

Dear Review,

Thank you for your fast answer and the very professionaly review of my manuskript. The points you noticed have been edited. Thank you for your comments.

  1. Define BRVA and CRT in abstract and text before use as well as any other abbreviations.

Response: BRVA, ERM, ILM, CRT were defined at the first mention.

  1. In abstract add logmar to brva. Also the postoperative vision is different in abstract and text.

Response: It has been corrected.

  1. Add the SD of BRVA in text.

Response: It has been added in the text.

  1. Please explain further why was a diamond dusted scraper needed in two cases (more adherent more friable membrane etc?)

Response: In two cases there was more adherent membrane and we needed a diamond dusted scraper.

  1. Please add a video for one of the operations to review.

Response: I can't download the video here (only PDF/ Word). The video will be emailed to mrs. Wang (Assitant Editor) and she can forward you.

(Wang: Please find the video by https://wetransfer.com/downloads/48374973af153d1c9a9fdd0a0ac322f520220919144606/c429a9e1dafc43f10687a5f485f3ad7020220919144627/4ea75c)

  1. Was there any post op complications such as retinal detachment or recurrent ERM?

Response: We don’t have any postoperative complications.

  1. In disclosure section you have to mention the disclosures of each author separately.

Response: It has been added in the text.

  1. In discussion you mention that the number of grasps to remove the whole ERM by Mylonas et al was reported as 71 ±35.2 compared to a mean of 4 grasps in your study but that was for initial mobilisation. This is not comparable. Please indicate that.

Response: We removed this not comparable discussion in manuscript.

  1. You have no conclusions in the conclusions section.

Response: conclusion has also been added

best regards 

Agharza Ashurov

Round 2

Reviewer 1 Report

Ashurov et al present a study of 23 eyes who underwent membrane peeling for various indications using the sharkskin forceps from Alcon, who the authors receive personal fees from. The authors find that these 23 patients have improved BRVA (20/60 to 20/40 in Snellen) and improved retinal thickness after surgery. On iOCT analysis, their cohort underwent an average of 4 initial grasps causing subretinal hyporeflectance in 17% of patients, retinal hemorrhage in 21%, and 0 retinal breaks. The authors addressed my major concern about their conclusions. In their response, they addressed a number of my concerns but did not amend their manuscript appropriately.

Remaining Comments:

1. The statistics still require revision. When comparing before peeling and after peeling, the appropriate test is a paired t-test between two groups if the data is parametric. The response states that they have some data for 3 months and all data for 6 months. I would recommend that they only report 6 month data for OCT CRT and BRVA and do a paired t-test. This should be clearly described in the methods (not just the response to the reviewer), and reported in the results (results only has t-test for OCT but not vision).

2. I still cannot review the high quality images for Fig 4-5. Currently, the figures basically show nothing in PDF form. I cannot review them and will send a note to the editor because this may not be the authors fault.

3. Please add this is a retrospective study to the methods section, not just the author response.

4. Please add single surgery macular hole closure success rate to the results.

Author Response

Dear Review,

Thank you for your fast answer and the very professionaly review of my manuskript. The points you noticed have been edited. Thank you for your comments.

  1. The statistics still require revision. When comparing before peeling and after peeling, the appropriate test is a paired t-test between two groups if the data is parametric. The response states that they have some data for 3 months and all data for 6 months. I would recommend that they only report 6 month data for OCT CRT and BRVA and do a paired t-test. This should be clearly described in the methods (not just the response to the reviewer), and reported in the results (results only has t-test for OCT but not vision).

Response:

  1. We added this in manuskript.

  1. I still cannot review the high quality images for Fig 4-5. Currently, the figures basically show nothing in PDF form. I cannot review them and will send a note to the editor because this may not be the authors fault.

  1. Please add this is a retrospective study to the methods section, not just the author response.

Response:

  1. We added this in manuskript.

  1. Please add single surgery macular hole closure success rate to the results.

Response:

  1. We added this in manuskript. ((The macular hole were closed successfully in all 7 eyes with a single surgery.))

best regards 

Agharza Ashurov

Reviewer 2 Report

Thank you for your revision.

Author Response

Dear reviewer,

attached you will find the revised manuscript. Thank you very much.

Best regards

A. Ashurov